# Two New Nonaborates with {B_9_} Cluster Open-Frameworks and Short Cutoff Edges

**DOI:** 10.3390/molecules27165279

**Published:** 2022-08-18

**Authors:** Chong-An Chen, Guo-Yu Yang

**Affiliations:** MOE Key Laboratory of Cluster Science, School of Chemistry and Chemical Engineering, Beijing Institute of Technology, Beijing 100081, China

**Keywords:** metal borates, oxoboron cluster, solvothermal syntheses, short cutoff edges

## Abstract

Two new nonaborates, Na_2_Ba_0.5_[B_9_O_15_]·H_2_O (**1**) and Na_4_Ca_1.5_[B_9_O_16_(OH)_2_] (**2**), have been solvothermally made with mixed alkali and alkaline-earth metal cationic templates. **1** and **2** are constructed by two different types of nonaborate clusters. In **1**, the [B_9_O_19_]^11−^ cluster is composed of three corner-sharing [B_3_O_7_]^5−^ clusters, of which two of them interconnect to the 1D B_3_O_7_-based chains and are further bridged to the 3D framework with 7 types of 10-MR channels by another [B_3_O_7_]^5−^ bridging cluster. The [B_9_O_18_(OH)_2_]^11−^ cluster in **2** is made of four BO_4_-sharing [B_3_O_8_]^7−^ clusters. As a 4-connected node, the interconnections of [B_9_O_18_(OH)_2_]^11−^ construct the unpreceded 2D layer with large 14-MR windows, which are further joined by H-bonds to the 3D supramolecular framework. UV–Vis absorption spectra reveal that both **1** and **2** have short cutoff edges below 190 nm, exhibiting bandgaps of 6.31 and 6.39 eV, respectively, indicating their potential applications in deep UV (DUV) regions.

## 1. Introduction

Borates have long been the research focus of inorganic crystalline materials due to their mineralogical and industrial importance since last century [1,2,3,4]. In the past three decades, scientists have not only been dedicated to enriching borates’ abundant structural diversity, but also explore their applications in fluorescence, catalysis, and nonlinear optics (NLO) [5,6,7,8,9]. As an oxyphilic element, boron atom can be triangular and tetrahedral coordinated with oxygen atoms, forming the fundamental building units of BO_3_ and BO_4_. These two units can further polymerize through corner-/edge-sharing to produce series of oxoboron clusters with different numbers of B atoms and configurations [10,11], which contribute to the abundant structural diversities of borates. To date, most of the commonly seen monomeric oxoboron clusters have boron atoms ranging from 3 to 7 [12,13]. Larger oxoboron clusters are usually made by the combinations of these four main types of clusters and BO_3/4_ units. For example, [B_8_O_14_(OH)_2_]^6−^ is made of the corner-sharing [B_5_O_9_(OH)]^4−^ and [B_3_O_6_(OH)]^4−^ cluster [14]; [B_9_O_18_(OH)_2_]^11−^ is formed by the corner-sharing [B_5_O_10_(OH)_2_]^7−^ and [B_4_O_9_]^6−^ cluster [15]; larger [B_20_O_32_(OH)_8_]^12−^ is built by 4 [B_4_O_7_(OH)_2_]^4−^ and 4 BO_3_^3−^ triangles [16,17,18]; while the largest oxoboron cluster [B_69_O_108_(OH)_18_]^27−^ is constructed by 3 [B_5_O_12_]^9−^, 12 [B_3_O_6_(OH)]^4−^, 6 BO_3_^3−^, 6 [BO_2_(OH)]^2−^, and 6 BO_4_^5−^ [10]. The different combinations and linkages of these cluster units contribute to the different configurations of the oxoboron clusters and their extended structures [6,7]. Due to the steric hindrances and polyhydroxyl characters, larger oxoboron clusters commonly generate lower dimensional structures without metal nodes and other bridging groups (clusters, organic amines, and transition metal complexes) [19,20,21].

In the past decades, large efforts have been concentrated on the alkali and alkaline-earth metal borates, of which the empty *d* and *f* orbitals in alkali-metal and alkaline-earth metal facilitate the transmittance of UV or even deep UV (DUV) light. Moreover, their asymmetric coordination geometries can reduce the symmetrical influences from templates to acentric oxoboron frameworks, enhancing the possibilities of acentric structures [22]. Large numbers of UV/DUV NLO borates have been obtained with alkali and alkaline-earth metal under different synthetic conditions, including the typical β-BaB_2_O_4_ (BBO), LiB_3_O_5_ (LBO), and CsLiB_6_O_10_ (CLBO) [23,24,25], and the newly obtained LiBa_3_(OH)-[B_9_O_16_][B(OH)_4_] [26], Li_2_CsB_7_O_10_(OH)_4_ [27], and A_10_B_13_O_15_F_19_ (A = K and Rb) [28]. From the view of the structure–property relationship, mixed alkali and alkaline-earth metal templates with different cationic radium are more likely to produce borates with compact structures, corresponding to the larger Second Harmonic Generation (SHG) response than single metal template, which are due to their different effects on the packing modes of anionic cluster originated from their different sizes and charges. Based on these considerations, our group has been working on the syntheses, structures, and properties of alkali and alkaline-earth metal borates under hydro(solvo)thermal conditions.

In this work, by using mixed alkali- and alkaline-earth metal templates, we successfully synthesized two new borates containing two different types of nonaborate clusters, namely Na_2_Ba_0.5_[B_9_O_15_]·H_2_O (**1**) and Na_4_Ca_1.5_[B_9_O_16_(OH)_2_] (**2**). In **1**, [B_9_O_19_]^11−^ cluster fundamental building blocks (FBBs) [29,30] are composed of three corner-sharing [B_3_O_7_]^5−^ rings (Figure 1a), of which two of them construct the independent 1D B_3_O_7_-based chains with opposite orientations, and are further bridged by another [B_3_O_7_]^5−^ bridging cluster to the 3D frameworks with seven types of intercommunicated 10-MR channels. In **2**, four BO_4_-sharing [B_3_O_8_]^7−^ clusters form the [B_9_O_18_(OH)_2_]^11−^ FBB (Figure 1b), which further construct the 2D unpreceded layer with large 14-MR windows. Notably, it is the first layered structure built by nonaborate cluster. H-bonds joined the layers to the 3D supramolecular open framework with three types of channels. UV–Vis absorption spectra show that both **1** and **2** have short cutoff edges below 190 nm, corresponding to bandgaps of 6.31 and 6.39 eV, respectively, indicating their potential applications in DUV regions.

## 2. Experimental Section

### 2.1. Syntheses

#### 2.1.1. Syntheses of **1**

A powder mixture of H_3_BO_3_ (0.248 g, 4.0 mmol), Na_2_[(HO)_2_B(O_2_)_2_B(OH)_2_]·6H_2_O, (sodium perborate, 0.064 g, 0.4 mmol), and Ba(OH)_2_·8H_2_O (0.315 g, 1.0 mmol) was added into 1 mL H_2_O and 2 mL pyridine. After stirring for 1 h, the resulting solution was sealed in a 25 mL Teflon-lined stainless-steel autoclave and heated at 230 °C for 7 days. After cooling down to room temperature spontaneously and being washed by distilled water, flake-like crystals were obtained in the yield of 37% (based on sodium perborate).

#### 2.1.2. Syntheses of **2**

A powder mixture of Na_2_[B_4_O_5_(OH)_4_]·8H_2_O (borax, 0.380 g, 1.0 mmol) and Ca(OH)_2_ (0.074 g, 1.0 mmol) was added into 4 mL H_2_O and 2 mL pyridine with constant stirring for 1 h. **2** was obtained under the same reaction temperature, time, and procedures as **1**. (Yield: 27% based on borax).

### 2.2. X-ray Crystallography

The single crystal diffraction data of **1** and **2** were collected on a Gemini A Ultra CCD diffractometer with graphite monochromated Mo Kα (Λ = 0.71073 Å) radiation at 293(2) K. The structures were solved by direct methods and refined by the full-matrix least-squares fitting on F^2^ method with the SHELX-2008 program package [31]. Anisotropic displacement parameters were refined for all atomic sites except for hydrogen atoms. Basic crystallographic data and structural refinement data are listed in Table 1. Detailed crystallographic data have been deposited on the Cambridge Crystallographic Data Centre: CCDC 2191292 (for **1**) and 2191293 (for **2**). These data can be obtained free of charge via http://www.ccdc.cam.ac.uk/conts/retrieving.html (accessed on 15 August 2022) or from the Cambridge Crystallographic Data Centre, 12 Union Road, Cambridge CB2 1EZ, UK; Fax: +44-1223-336-033; or email: deposit@ccdc.cam.ac.uk.

### 2.3. General Procedure

All the reagents were analytical grade and used without any further purification. The powder X-ray diffraction (PXRD) patterns of the title compounds were both collected on a Bruker D8 Advance X-ray diffractometer (Bruker, Karlsruhe, Germany) with Cu Kα radiation (λ = 1.54056 Å), 2θ scanning from 5–50° with a step size of 0.02° at room temperature. UV–Vis absorption spectra were recorded on a Shimadzu UV3600 spectrometer (Shimadzu, Kyoto, Japan) with the wavelengths ranging from 190 to 800 nm. IR spectra were obtained on a Nicolet iS10 FT-IR spectrometer (Thermo Fisher Scientific, Waltham, MA, USA) with the wavenumbers ranging from 4000 to 400 cm^−1^. Thermogravimetric analyses were carried out on a Mettler Toledo TGA/DSC 1100 analyzer (Mettler Toledo, Zurich, Switzerland), heating up from 25 to 1000 °C with a heating rate of 10 °C/min under air atmosphere.

## 3. Result and Discussion

### 3.1. Structure of ***1***

Single crystal X-ray analyses show that **1** crystalizes in the monoclinic space group *P*2_1_/*c*. Its asymmetric unit contains a [B_9_O_19_]^11−^ cluster formed by three corner-sharing [B_3_O_7_]^5−^ clusters, 2 Na^+^, 0.5 Ba^2+^ and 1 water molecule (Figure 1a). In order to describe the construction of the structure, three corner-sharing [B_3_O_7_]^5−^ clusters are named in the shorthand notations of B_3_-I, B_3_-II, and B_3_-III. Each of these three B_3_ clusters is four-connected, bonding with two others in the same quantities (Figure 1b). B_3_-I and B_3_-II first interconnects to form the 1D [B_3_O_7_]_n_^5n−^ chain along the *b* axis (Figure 1c), exhibiting two opposite orientations with an *C*_2_-symmetry axis. These two different orientated 1D chains alternately arrange in -ABAB- and -AAA- sequences along *a*- and *c*-axes (Figure 1d), respectively, which are further bridged to 3D framework by the B_3_-III bridging cluster (Figure 1e–g). There exist seven different types of 10-MR channels in the 3D framework of **1** (Figure 1h,i). Along the *a* axis, 10-MR channels A and B show same shapes and delimitations (B5O_4_-B1O_3_-B2O_3_-B7O_4_-B6O_3_-B5O_4_-B1O_3_-B2O_3_-B7O_4_-B6O_3_) but different orientations, they alternately arrange along the *b* axis. Another 10-MR channel C with different shape is delineated by B5O_4_-B9O_3_-B8O_3_-B3O_4_-B4O_3_-B5O_4_-B9O_3_-B8O_3_-B3O_4_-B4O_3_. Channels D, E, and F appear along the *b* axis, of which channel D and F have the same delimitations as channel C, but showing different shapes, while channel E is made from the alternations of B7O_4_-B2O_3_-B3O_4_-B4O_3_-B6O_3_-B7O_4_-B2O_3_-B3O_4_-B4O_3_-B6O_3_. Channel G along the *c* axis has the same components as those in A and B. These seven types of channels make up the intercommunicated channel system of **1**. To the best of our knowledge, such complicated channel system is only found in the aluminoborate [H_3_O]K_3.52_Na_3.48_{Al_2_[B_7_O_13_(OH)][B_5_O_10_]-[B_3_O_5_]}[CO_3_] (Appendix A) [32], which exhibits the 3D porous layer with seven intercommunicated channel networks. Five-coordinated Na^1+^ and seven-coordinated Na^2+^ are located in the channels F, while twelve-coordinated Ba^2+^ are located in the channels E, compensating the negative charges of the frameworks (Appendix A). It is worth noticing that each of the two Na^1+^, two Na^2+^, and one Ba^2+^ alternate through corner-sharing, producing the rare [Na_4_BaO_28_] pentamers (Appendix A).

### 3.2. Structure of ***2***

Single crystal X-ray analyses show that **2** crystalizes in the monoclinic space group *C*2/*c*. Its asymmetric unit contains a [B_4.5_O_11_(OH)]^9.5−^ cluster unit, 2 independent Na^+^, and 0.75 Ca^2+^ (Figure 2a). In the [B_4.5_O_11_(OH)]^9.5−^ unit, the occupation of B2 is 0.5, leading to the [B_9_O_18_(OH)_2_]^11−^ FBB through symmetrical operation, which can be written as [9: 3Δ + 6T] (Christ and Clark descriptor) or 3Δ6□: Δ2□ − <Δ2□ > − <Δ2□> − <Δ2□> (Burns descriptor) [29,30]. According to the BVS calculations, the bond valences of the terminal O1 and O7 are 1.55 and 1.27, respectively. Meanwhile, there is no proper atomic site for H atom on O1, indicating that O1 is non-protonated and O7 is protonated.

In the structure, each [B_9_O_18_(OH)_2_]^11−^ FBB links with four same ones to generate the 2D layer with large 14-MR windows (Figure 2b,c). Adjacent layers are arranged in -ABAB- manners along the *c* axis and show a curtain extend of dislocation (Figure 2d,e). Interlayered H-bond interactions join adjacent layers to the 3D supramolecular open framework (Appendix A), containing three different types of supramolecular channels (Figure 2d,f). Charge-balancing cations all locate in the large 14-MR windows (Appendix A). Na^1+^ and Na^2+^ are 6- and 5-coordinated with Na-O bond length ranging from 2.312–2.695Å and 2.259–2.487Å, respectively, while Ca^2+^ is 8-coordinated, corresponding to the bond length range of 2.354–2.840 Å. Na^1+^, Na^2+^, and Ca^2+^ interconnect to construct the 3D metal-oxygen porous layer (Appendix A), which is rare in borates.

### 3.3. Structure Comparision

To date, there are three main types of nonaborate cluster FBBs with different configurations that have been obtained. According to the classifications and definitions of oxoboron clusters proposed by Christ, Clark and et al. [29,30], these nonaborate cluster FBBs can be divided into the following types:(1)[B_9_O_14_(OH)_4_]^5−^ and [B_9_O_18_(OH)_2_]^11−^ FBBs: These two types of FBBs are composed of two different {B_5_} and {B_4_} clusters, which produce the 1D chains and 3D framework with different types of guest templates [15,33]. In Ni(en)_3_·Hen·[B_9_O_13_(OH)_4_]·H_2_O, the linear alternations of [B_9_O_14_(OH)_4_]^5−^ ([9: (5: 4Δ + 1T) + (4: 2Δ + 2T)]) FBB build the 1D chains (Figure 3a) [33], while in Li_2_[B_4_O_7_][B_5_O_8_(OH)_2_], the [B_9_O_18_(OH)_2_]^11−^ ([9: (5: 2Δ + 3T) + (4: 2Δ + 2T)]) FBB construct the 3D diamond framework (Figure 3b) [15].(2)[B_9_O_19_]^11−^ FBB: This kind of [B_9_O_19_]^11−^ ([9: (6: 6T) + 3Δ]) FBB is composed of a planar [B_6_O_16_]^14−^ cluster (three edge-sharing [B_3_O_9_]^9−^ cluster) and three capping BO_3_ triangles. As a 6-connected node, it constructs series of 3D acs frameworks with large 21-MR channels (Figure 3c) [26,34,35].(3)[B_9_O_12_(OH)_6_]^3−^, [B_9_O_16_(OH)_3_]^8−^ and [B_9_O_16_(OH)_4_]^9−^ FBBs: There are two different configurations of these nonaborate cluster FBBs, including the three corner-sharing [B_3_O_8_]^7−^ clusters-made [B_9_O_19_]^11−^ ([9: 3×(3: 2Δ + T)], FBB in **1**) and the four BO_4_-sharing B_3_O_3_ rings-made [B_9_O_16_(OH)_2_]^7−^ ([9: 3Δ + 6T)]) and its derived forms (FBB in **2**). As described above, [B_9_O_19_]^11−^ make up the 3D framework with intercommunicated channel networks. For the [B_9_O_16_(OH)_2_]^7−^ and its derived clusters, only isolated and 1D structures were obtained before. [B_9_O_12_(OH)_6_]^3−^ ([9: 6Δ + 3T]) FBB in [C(NH_2_)_3_]_3_[B_9_O_12_(OH)_6_] is an isolated cluster [36], joined by the abundant H-bond interactions to 3D supramolecular frameworks (Figure 3d). [B_9_O_16_(OH)_3_]^8−^ ([9: 5Δ + 4T]) and [B_9_O_16_(OH)_4_]^9−^ ([9: 4Δ + 5T]) FBBs in Na_4_[B_9_O_14_(OH)_3_]·0.5H_2_O and Na_5_[B_9_O_14_(OH)_4_] features two different 1D belt-like structures [37], respectively, which are originated from configurations of the clusters. In Na_4_[B_9_O_14_(OH)_3_]·0.5H_2_O, the Z-shape [B_9_O_16_(OH)_3_]^8−^ interconnects via four terminal O atoms from the four [B_3_O_3_] rings, of which the terminal O atoms on the outboard [B_3_O_3_] ring is linked with those on the medial rings, making the 1D zigzag belt with 8-MR windows (Figure 3e), while in Na_5_[B_9_O_14_(OH)_4_], the half-occupied central B atom leads to the C-shape [B_9_O_16_(OH)_4_]^9−^, which further interconnects through the same linkages with the Z-shape [B_9_O_16_(OH)_3_]^8−^, constructing the 1D linear belt (Figure 3f). The 2D layer in **2** is the first 2D layer built from the nonaborate clusters.

**Figure 3 molecules-27-05279-f003:**
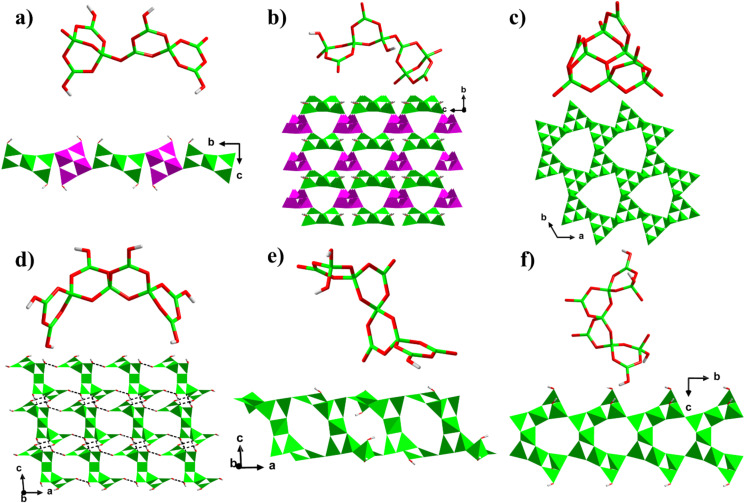
(**a**) [B_9_O_14_(OH)_4_]^5−^ FBB and the 1D chain in Ni(en)_3_·Hen·[B_9_O_13_(OH)_4_]·H_2_O. (**b**) [B_9_O_18_(OH)_2_]^11−^ FBB and the 3D diamond framework in Li_2_[B_4_O_7_][B_5_O_8_(OH)_2_]. (**c**) [B_9_O_19_]^11−^ FBB and the 3D acs framework in LiBa_3_(OH)[B_9_O_16_][B(OH)_4_]. (**d**) [B_9_O_12_(OH)_6_]^3−^ FBB and the 3D supramolecular framework in [C(NH_2_)_3_]_3_[B_9_O_12_(OH)_6_]. (**e**) [B_9_O_16_(OH)_3_]^8−^ FBB and the 1D zigzag belt in Na_4_[B_9_O_14_(OH)_3_]·0.5H_2_O. (**f**) [B_9_O_16_(OH)_4_]^9−^ FBB and the 1D linear belt in Na_5_[B_9_O_14_(OH)_4_].

### 3.4. Powder XRD Patterns

The experimental PXRD patterns of **1** and **2** were all in good agreement with the simulated patterns obtained from single crystal data, which confirm the purities of the samples. The inconsistency of diffraction peaks’ intensities between experimental and simulated patterns is attributed to the various preferred orientations of the samples (Appendix A).

### 3.5. IR Spectra

The absorption bands and peaks in the IR spectra of **1** and **2** are similar in the range of 4000–400 cm^−1^. Only **1** is discussed in detailed. The wide absorption bands from 3663 to 2985 cm^−1^ are assigned to the stretching vibrations of -OH groups. The absorption peaks at 1624 cm^−1^ are the vibrations of H-O-H. The peaks ranging from 1509–1290 cm^−1^ and 1146–970 cm^−1^ are attributed to the B–O asymmetrical stretching of the BO_3_ and BO_4_ units, respectively. The sharp peaks at 941 and 827 cm^−1^ are the symmetric stretching vibrations of BO_3_ and BO_4_ units, respectively, while the bending vibrations of these units appear in the range of 749–601 cm^−1^ (Appendix A).

### 3.6. UV–Vis Absorption Spectra

UV–Vis absorption spectra of **1** and **2** were obtained in the wavelength range of 190–800 nm. As shown in Figure 4, both **1** and **2** have the short cutoff edges below 190 nm, showing the experimental band gaps of 6.31 eV (**1**, Figure 4a) and 6.39 eV (**2**, Figure 4b), respectively, indicating their potential applications in DUV regions. The short DUV cutoff edges and large bandgaps of **1** and **2** are mainly originated from the absences of *d*-*d* and *f*-*f* electron transitions of alkali- and alkaline-earth metal cations, which have been proved in other borates with DUV cutoff edges, including Li_2_CsB_7_O_10_(OH)_4_ (6.35 eV) [27] and CsB_7_O_10_(OH)_2_ (6.60 eV) [38]. No such cases were found in other borates without alkali and alkaline-earth metal cations.

### 3.7. Thermal Analysis

The thermal stabilities of **1** and **2** were examined in 25–1000 °C with the heating rate of 10 °C/min under the air atmosphere (Appendix A). For **1**, there was only one step weight loss of 4.49% (Cal: 3.83%) from 122 to 311 °C, corresponding to the removal of one lattice water molecule. For **2**, the 3.64% (Cal: 3.33%) weight loss from 387 to 676 °C is assigned to one water molecule from the dehydration of two -OH groups.

## 4. Conclusions

In summary, two new nonaborates templated by mixed alkali- and alkaline earth metal cations, Na_2_Ba_0.5_[B_9_O_15_]·H_2_O (**1**) and Na_4_Ca_1.5_[B_9_O_16_(OH)_2_] (**2**), were successfully made under solvothermal conditions. **1** is made of three different [B_3_O_7_]^5−^ clusters, of which two of them interconnects to make the 1D chains that are further joined to 3D framework by another [B_3_O_7_]^5−^ clusters. There exist seven different types of intercommunicated channels in the framework of **1**, which is first observed in 3D B-O frameworks. The nonaborate cluster in **2** is composed of four BO_4_-sharing [B_3_O_8_]^7−^ clusters. It constructs the 2D monolayer with large 14-MR windows stacking in -ABAB- sequence, filling the blank of 2D layered nonaborates. H-bonds link adjacent layers to the 3D supramolecular open framework with three types of supramolecular channels. UV–Vis absorption spectra reveal that both **1** and **2** exhibit the short DUV cutoff edges below 190 nm, and the bandgaps are 6.31 and 6.39 eV, respectively, indicating that they have potential applications in DUV regions. Further work on exploring new DUV borates with excellent physical chemical properties is underway by using mixed alkali- and alkaline earth metal cationic templates under hydro(solvo)thermal conditions.

## Data Availability

Not applicable.

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
