# Peer review of "Two New Nonaborates with {B9} Cluster Open-Frameworks and Short Cutoff Edges"

_molecules, 2022, doi:10.3390/molecules27165279_

Round 1

Reviewer 1 Report

The authors discussed Two New Nonaborates with {B9} Cluster Open-Frameworks 2 and Short Cutoff Edges which look very interesting with minimal comments.

Experimental Section

Ø  General Procedure What is in this section should be the last report in this section.

Ø  Syntheses of 1. Give the full detail of the 1 and repeat it in the others. you should only identify it by using letters.

Author Response

Response: We sincerely appreciate your valuable suggestions. We have revised this section according to your suggestions.

Reviewer 2 Report

This is an interesting manuscript which reports some good science that is worthy of publication in due course.  As a borate chemist I can appreciate  the structural aspects and  DUV properties of the new materials but I have some major issues with the presentation of the work.  In its present form I do not think it will be understandable to the more general reader and should not be published.

The first paragraph of the introduction needs to be rewritten with careful thought for the precision of the chemical language used.

 e.g. L28 ‘independent oxoboron clusters’ –  two points here:

- meaning of independent? (isolated, insular?)

- these are not oxoboron clusters.  They are anionic oxidoborate giant  lattices (1)  or anionic hydroxidooxidoborate giant lattices  (2)  and should be formulated as such so as to make sense.

Compound 1 is Na2Ba0.5[B9O15].H2O where there is a FBB (framework building block) of [B9O15]n3n- or more simply [B9O15]3-.

 All the formula in this paragraph need charges associated with them to make any sense.

Compound 2 is correctly formulated and contains the anionic giant lattice [B9O16(OH)2]7-

[B9O16(OH)2]n7n-

L62 the term structural building unit SBU is used  This completely ignores the well-used FBB system developed by  Christ and Clark  (Phys Chem Miner 1977, 2, 59-87) and  Grice, Burns Hawthorne ( Can Min 1999, 37, 731. )  This section needs rewriting  to use the familiar terms that have been around in the west for more  than 50 years. 

Experimental  - it would be good to add the name sodium perborate and borax, as well as the formula,  into the synthetic sections.  Again structural precision is absent in the formula. Borax is an insular hydrated salt Na2[B4O5(OH)4].8H2O and the perborate is an insular hydrated salt Na2[(HO)2B(O2)2B(OH)2].6H2O.

Discussion of structures – use FBB language  and not cluster language.

Line drawings of the repeating units would also be appreciated as separate Figures.

Likewise SBU need changing to FBB in Figures.

The manuscript has plenty of modern day citations from Chinese scientists  but ignores some seminal work from the west undertaken in the last century.  A  through background literature search is needed for the re-write.

Author Response

The responses to the Reviewers’ comments:

Reviewer: 2

Comments:

This is an interesting manuscript which reports some good science that is worthy of publication in due course. As a borate chemist I can appreciate the structural aspects and DUV properties of the new materials but I have some major issues with the presentation of the work. In its present form I do not think it will be understandable to the more general reader and should not be published.

  1. The first paragraph of the introduction needs to be rewritten with careful thought for the precision of the chemical language used.

 e.g. L28 ‘independent oxoboron clusters’ – two points here:

- meaning of independent? (Isolated, insular?)

- these are not oxoboron clusters.  They are anionic oxidoborate giant lattices (1) or anionic hydroxidooxidoborate giant lattices (2) and should be formulated as such so as to make sense.

Compound 1 is Na2Ba0.5[B9O15]‧H2O where there is a FBB (framework building block) of [B9O15]n3n- or more simply [B9O15]3-.

Compound 2 is correctly formulated and contains the anionic giant lattice [B9O16(OH)2]7- [B9O16(OH)2]n7n-

   All the formula in this paragraph needs charges associated with them to make any sense.

Response: We sincerely appreciate your valuable suggestions. ‘Independent oxoboron clusters’ means monomeric clusters, not those composed of two different monomeric clusters. We have rewritten this part for better clarity. Besides, formulae in this paragraph have been revised with anionic giant lattices.

  1. L62 the term structural building unit SBU is used. This completely ignores the well-used FBB system developed by Christ and Clark (Phys Chem Miner 1977, 2, 59-87) and Grice, Burns Hawthorne (Can Min 1999, 37, 731.) This section needs rewriting to use the familiar terms that have been around in the west for more than 50 years.

Response: We sincerely appreciate your valuable suggestions. We have used the well-used FBB system proposed by Christ, Clark and Burns to replace the SBU descriptions in each part of the revised manuscript.

  1. Experimental- it would be good to add the name sodium perborate and borax, as well as the formula, into the synthetic sections. Again, structural precision is absent in the formula. Borax is an insular hydrated salt Na2[B4O5(OH)4]‧8H2O and the perborate is an insular hydrated salt Na2[(HO)2B(O2)2B(OH)2]‧6H2O.

Response: We sincerely appreciate your valuable suggestions. In this section, we have added the name borax and sodium perborate with their precise formulae Na2[B4O5(OH)4]‧8H2O and Na2[(HO)2B(O2)2B(OH)2]‧6H2O, respectively.

  1. Discussion of structures–use FBB language and not cluster language.

Line drawings of the repeating units would also be appreciated as separate Figures.

Likewise, SBU need changing to FBB in Figures.

Response: We sincerely appreciate your valuable suggestions. In the structural description and comparison sections, cluster language has been replaced with FBB language. Line drawings of the repeating units of the compounds 1 and 2 have been supplied. Figures and their captions have also been changed from SBU to FBB.

  1. The manuscript has plenty of modern-day citations from Chinese scientists but ignores some seminal work from the west undertaken in the last century. A thorough background literature search is needed for the re-write.

 Response: We sincerely appreciate your valuable suggestions. We have revised the first paragraph and added series of references that represent some seminal work from the west in the last century.

Round 2

Reviewer 2 Report

no additional comments